# Development of Anti-OSCAR Antibodies for the Treatment of Osteoarthritis

**DOI:** 10.3390/biomedicines11102844

**Published:** 2023-10-19

**Authors:** Gyeong Min Kim, Doo Ri Park, Thi Thu Ha Nguyen, Jiseon Kim, Jihee Kim, Myung-Ho Sohn, Won-Kyu Lee, Soo Young Lee, Hyunbo Shim

**Affiliations:** 1Department of Life Sciences, Ewha Womans University, Seoul 03760, Republic of Korea; wwwwvvv@nate.com (G.M.K.); pdr315@naver.com (D.R.P.); hanguyen15490@gmail.com (T.T.H.N.); abcdefg108@naver.com (J.K.); maydate@naver.com (J.K.); 2The Research Center for Cellular Homeostasis, Ewha Womans University, Seoul 03760, Republic of Korea; 3New Drug Development Center, Osong Medical Innovation Foundation, Cheongju 28160, Republic of Korea; sohnho2@kbiohealth.kr (M.-H.S.); gre7@kbiohealth.kr (W.-K.L.)

**Keywords:** OSCAR, osteoarthritis, human antibody, chondrocyte, disease-modifying osteoarthritis drugs, DMOAD

## Abstract

Osteoarthritis (OA) is the most common joint disease that causes local inflammation and pain, significantly reducing the quality of life and normal social activities of patients. Currently, there are no disease-modifying OA drugs (DMOADs) available, and treatment relies on pain relief agents or arthroplasty. To address this significant unmet medical need, we aimed to develop monoclonal antibodies that can block the osteoclast-associated receptor (OSCAR). Our recent study has revealed the importance of OSCAR in OA pathogenesis as a novel catabolic regulator that induces chondrocyte apoptosis and accelerates articular cartilage destruction. It was also shown that blocking OSCAR with a soluble OSCAR decoy receptor ameliorated OA in animal models. In this study, OSCAR-neutralizing monoclonal antibodies were isolated and optimized by phage display. These antibodies bind to and directly neutralize OSCAR, unlike the decoy receptor, which binds to the ubiquitously expressed collagen and may result in reduced efficacy or deleterious off-target effects. The DMOAD potential of the anti-OSCAR antibodies was assessed with in vitro cell-based assays and an in vivo OA model. The results demonstrated that the anti-OSCAR antibodies significantly reduced cartilage destruction and other OA signs, such as subchondral bone plate sclerosis and loss of hyaline cartilage. Hence, blocking OSCAR with a monoclonal antibody could be a promising treatment strategy for OA.

## 1. Introduction

Osteoarthritis (OA) is the most common degenerative joint disease with accompanying inflammation and pain that can significantly impair patient quality of life. It is characterized by articular cartilage destruction, which is caused by dysregulation of the articular chondrocytes [1,2]. These cells are exposed continuously to external mechanical forces and play key roles in cartilage homeostasis because they produce both the cartilage extracellular matrix (ECM) molecules and the metalloproteinases (MMPs) that degrade them [3,4]. In OA, this anabolic/catabolic balance is tipped towards catabolism, cartilage degradation, chondrocyte apoptosis, and lacunar emptying [5,6]. Additionally, as a whole-joint disease, OA elicits damage to other joint tissues, including subchondral bone, meniscus, synovial membrane, ligaments, capsule, and infrapatellar fat pad [2,7]. The non-cartilage tissues and cells around the joint also participate in OA pathogenesis. In particular, OA is associated with increased numbers of immune cells in the synovium that release pro-inflammatory cytokines such as interleukin (IL)-1β, IL-6, and tumor necrosis factor (TNF)-α: These factors promote the cartilage catabolic functions of chondrocytes while simultaneously downregulating their cartilage anabolism [8]. Moreover, the pro-inflammatory factors also activate the osteoclasts and osteoblasts in the subchondral bone underneath the cartilage, which causes bone sclerosis that accelerates cartilage degradation [9,10]. In addition to these cellular and biochemical factors, OA pathogenesis is also accompanied and affected by the biomechanical changes in chondrocytes and the articular cartilage [7,11].

Despite numerous studies on OA pathogenesis and extensive efforts to develop effective treatments for OA, there are currently no approved disease-modifying OA drugs (DMOADs) [12,13], and OA treatment largely relies on pain relief agents or arthroplasty. In our recent study, we proposed that osteoclast-associated receptor (OSCAR) might be an effective DMOAD target [14]. OSCAR is a member of the leukocyte receptor complex (LRC)-encoded protein family characterized by the presence of extracellular immunoglobulin (Ig)-like domains [15,16] and recognizes specific sequences in type I, II, and III collagens [17,18,19]. While OSCAR was first identified as a co-stimulatory receptor that is required for complete osteoclast differentiation [20], it is now clear that it is also expressed by other myeloid lineage cells, including monocytes, macrophages, neutrophils, and monocyte-derived dendritic cells. Moreover, it is associated closely with bone diseases [16,21,22]. Although articular chondrocytes usually do not express OSCAR or do so only at low levels, its expression is strongly upregulated in OA [14]. Our study also showed that when OSCAR was knocked out, mice that underwent OA-inducing destabilization of the medial meniscus (DMM) surgery demonstrated significantly less articular cartilage destruction. In vitro experiments then showed that OSCAR promotes OA by downregulating osteoprotegerin (OPG) and upregulating TNF-related apoptosis-inducing ligand (TRAIL), which in turn induces chondrocyte apoptosis. Notably, OA in mice was ameliorated by intra-articular injections of soluble human OSCAR-Fc fusion protein, which binds to murine as well as human collagen and acts as a decoy receptor [14]. Thus, OSCAR plays a vital role in OA progression and is a potential therapeutic target [22,23].

In the present study, we aimed to develop anti-OSCAR antibodies and demonstrate their capacity to block OSCAR activities, potentially slowing the progression of OA and ameliorating its symptoms. To achieve this goal, OSCAR-neutralizing antibodies were generated and optimized by phage display, and their efficacies were tested in vitro and in vivo. It is anticipated that the results from the study confirm the crucial role of OSCAR in OA pathogenesis, as well as provide a novel therapeutic approach to the treatment of OA.

## 2. Materials and Methods

### 2.1. Mice

Primary articular chondrocytes were obtained from Institute of Cancer Research (ICR) mice (see Section 2.7). Male C57BL/6J mice (9–10-week-old, Japan SLC, Inc., Hamamatsu, Japan) were used for the experimental OA studies. All mice were housed at 5 or fewer per cage in pathogen-free barrier facilities. The maintenance conditions were: temperature at 24–26 °C, humidity at 30–60%, and 12 h light/dark cycles. After one week of adaptation in the animal house, the OA-experiment mice were randomly allocated to each experimental group (*n* = 6 each). All animal experiments were approved by the Institutional Animal Care and Use Committees (IACUC) (Protocol No: IACUC 23-016) of Ewha Womans University and followed National Research Council Guidelines. Mice experiments were conducted following the ARRIVE guidelines.

### 2.2. Production of OSCAR-Fc Fusion Proteins

FreeStyle™ 293F cells (Thermo Fisher Scientific, Waltham, MA, USA) were transfected with pVITRO1 vector (Invivogen, San Diego, CA, USA) harboring a gene encoding the extracellular domains (amino acids 19–233) of human OSCAR or the corresponding domains of mOSCAR-Fc fused to the Fc region of human IgG1 (hOSCAR-Fc and mOSCAR-Fc, respectively). The secreted OSCAR-Fc proteins were loaded onto Thermo Scientific™ Pierce™ Protein G-Sepharose bead columns (Thermo Fisher Scientific). The columns were washed twice with 5 column volumes of PBS, and the bead-bound proteins were eluted with an elution buffer (100 mM glycine, pH 2.0) and neutralized with 0.5 volume of 1 M Tris-HCl (pH 7.0). The neutralized proteins were then extensively dialyzed against PBS and kept frozen at −80 °C.

### 2.3. Identification of OSCAR-Binding Antibodies

Anti-OSCAR antibodies were isolated from synthetic human single-chain variable fragment (scFv) libraries by phage display biopanning and subsequent ELISA screening that were conducted according to previously described protocols [24,25]. The libraries were panned in alternating rounds against hOSCAR-Fc and mOSCAR-Fc. To minimize the enrichment of Fc-binding antibodies, human IgG1 (hIgG) was added at 100 μg/mL as a competitor during the binding step. After three to four rounds of panning, the output clones were screened by ELISA for binding to hOSCAR-Fc, mOSCAR-Fc, or hIgG1 as an Fc control.

### 2.4. Production of Anti-OSCAR mAbs

The VH and VL domains of the anti-OSCAR scFv antibodies were separately cloned into the pcIW3.3 vector. The ExpiCHO™ cell line (Thermo Fisher Scientific) was transfected with the expression constructs in a 2:1 light chain:heavy chain ratio, after which the transfected cells were cultured for 8–10 days according to the manufacturer’s recommended protocol. The secreted IgGs were purified as described above for OSCAR-Fc.

### 2.5. Chondrocyte-Based Assay to Identify Antibodies That Inhibit OSCAR In Vitro

The mouse chondrogenic cell line ATDC5 was cultured in Dulbecco’s Modified Eagle Medium (DMEM)/F-12 1:1 (sh30023.01; Hyclone, Inc., Logan, UT, USA) supplemented with 5% fetal bovine serum and 1% penicillin/streptomycin at 37 °C in a humidified atmosphere with 5% CO_2_. The ATDC5 cells were plated in a 6-well culture plate at a density of 2 × 10^5^ cells/well. The next day, anti-OSCAR Abs (20 μg/mL), the hOSCAR-Fc (20 μg/mL) positive control, or the hIgG negative control (20 μg/mL) were added for 30 min. Collagen (Sigma-Aldrich, St. Louis, MO, USA; 5 μg/mL) was subsequently added, and the cells were incubated for another 24 h. Quantitative RT-PCR was then conducted to determine the expression of *Oscar*, *Epas1*, *Mmp3*, and *Mmp13*.

### 2.6. RNA Isolation and Quantitative RT-PCR

Total RNAs were isolated from ATDC5 or primary chondrocytes by using TRIzol reagent (Invitrogen, Carlsbad, CA, USA) and reverse-transcribed with the Superscript cDNA synthesis kit (Invitrogen) to generate cDNA. Real-time PCR was performed using the KAPA SYBR Green fast qPCR kit (Kapa Biosystems, Inc., Wilmington, MA, USA) on a Step One Plus real-time PCR machine (Applied Biosystems, Foster City, CA, USA). The samples were analyzed in triplicate using 2^−ΔΔCT^ method, and the data were normalized to β-actin (*Actb*) mRNA expression. Table 1 shows the primer sequences.

### 2.7. Primary Articular Chondrocyte Assay to Identify Antibodies That Inhibit OSCAR In Vitro

Articular chondrocytes were isolated from the femoral condyles and tibial plateaus of 4–5-day-old ICR mice by digestion with 0.2% collagenase type II [26]. The chondrocytes were maintained in Dulbecco’s modified Eagle’s medium (DMEM; HyClone) containing 10% fetal bovine serum. An OSCAR-binding triple helical peptide [OSC; GPC-(GPP)_5_-GPOGPAGFO-(GPP)_5_-GPC-amide where O denotes 4-hydroxyproline] [19] and the GPP_10_ collagen-mimetic negative control peptide [GPC-(GPP)_10_-GPC-amide] (CAS No. 2260816-94-0) were purchased from University of Cambridge. The purity of the peptides was analyzed by high-performance liquid chromatography. Chondrocytes were incubated with OSC or GPP_10_ (10 μg/mL) in the presence or absence of anti-OSCAR antibodies (10 μg/mL), the hOSCAR-Fc positive control (10 μg/mL), or the hIgG negative control (10 μg/mL) for 2 h and treated with IL-1β (5 ng/mL; R&D Systems, Minneapolis, MN, USA) for 48 h. Quantitative RT-PCR was then conducted to determine the expression of *Oscar* and *Tnfsf10*. Moreover, articular chondrocyte apoptosis was determined by using the Terminal deoxynucleotidyl transferase dUTP nick-end labeling (TUNEL) assay using a kit (Apoptosis Detection Kit, Lot #2397029, Millipore, Temecula, CA, USA).

### 2.8. Osteoclast Differentiation Assay to Identify Antibodies That Inhibit OSCAR In Vitro

Bone marrow-derived macrophages (BMMs) were prepared from mice as previously described [27]. Primary osteoblasts (pOBs) were isolated from the calvariae of newborn mice by digestion with 0.8 unit/mL dispase (Roche Applied Science, Penzberg, Germany) and 0.1% collagenase (Sigma-Aldrich). BMMs and pOBs are cultured in alpha-minimum essential medium (α-MEM; HyClone) containing 10% fetal bovine serum (HyClone) and 1% penicillin/streptomycin (Gibco, Billings, MT, USA). For co-culture experiments, BMMs (2 × 10^5^ cells/well) and calvarial osteoblasts (2 × 10^4^ cells/well) were plated on 48-well culture plates and cultured in medium containing 1 μM prostaglandin E2 (Sigma-Aldrich), 10 nM 1,25-(OH)_2_ vitamin D3 (Sigma-Aldrich), and various concentrations of anti-OSCAR antibodies, the hOSCAR-Fc positive control, or the hIgG negative control for 6 days [28]. The cells were then fixed and stained for TRAP activity (Sigma-Aldrich). The number of cells with more than three nuclei (i.e., the mature osteoclasts) were counted.

### 2.9. Affinity Maturation of the B4 Clone by Light Chain Shuffling

The light chain repertoire of a synthetic human scFv library [25] was amplified by PCR using primers Uni-JH-f (5′-GGT ACA CTG GTG ACC GTG AGC-3′) and pC3X-b (5′-AAC CAT CGA TAG CAG CAC CG-3′). The variable heavy domain of the parental anti-OSCAR scFv clone B4 was amplified by using primers pC3X-f (5′-GCA CGA CAG GTT TCC CGA C-3′) and Uni-JH-b (5′-GCT CAC GGT CAC CAG TGT ACC-3′). The two DNA fragments were assembled by overlap extension PCR using primers pC3X-short-f (5′-TAT TGC CTA CGG CAG CC-3′) and p3CX-short-b (5′-ATC ACC GGA ACC AGA GC-3′). The PCR product was purified by agarose gel electrophoresis using QIAquick^®^ gel extraction kit (Qiagen GmbH, Hilden, Germany), digested with SfiI (New England Biolabs, Ipswitch, MA, USA), and ligated to SfiI-digested pComb3X phagemid vector. TG1 *E. coli* cells were transformed with the ligated DNA by electroporation and grown overnight at 37 °C on an LB-agar plate with 100 μg/mL ampicillin and 2% (*w*/*v*) glucose. The phage library was rescued from the overnight culture as described previously [29] and panned against hOSCAR-Fc over three rounds. For the first and second rounds of panning, 1 μg/mL of the antigen was used for immunotube coating, and the tube was washed 5 times with PBST (PBS with 0.05% Tween-20). For the third round, 0.2 μg/mL of the antigen was used, and the wash cycle consisted of 8 normal washes and 2 extended washes (30 min incubation) with PBST. After the panning, the output clones were screened by ELISA, as described above.

### 2.10. Affinity Maturation of the D11 Clone by CDR-H3 Randomization

The overlapping 2-amino acid residues of the complementarity-determining region (CDR)-H3 sequence of the anti-OSCAR scFv D11 were randomized by using the NNK degenerate codon. Oligonucleotides encoding the CDR-H3 sequence with randomization were inserted into the D11 sequence by overlap extension PCR. SfiI digestion, ligation/transformation, and the subsequent panning and screening steps were performed as described above. 

### 2.11. Measurement of Antibody Affinity by Biolayer Interferometry (BLI)

The binding kinetics of B4, D11, and the affinity-matured clones were analyzed by biolayer interferometry using the Octet^®^ instrument (Fortebio, Freemont, CA, USA). IgG antibodies (5 μg/mL) were captured on the FAB2G sensor, and their binding to 4.69–75 nM (in two-fold serial dilution) of hOSCAR-Fc or mOSCAR-Fc in PBST binding buffer was measured. The data were fitted to a 1:1 Langmuir binding model to estimate *k*_on_, *k*_off_, and *K*_D_ values.

### 2.12. Experimental OA in Mice

Experimental OA was induced in 10–12-week-old mice by DMM surgery. DMM surgery [30,31] involves surgically removing the medial meniscus ligament from the right knee joint of the hind limb. The mice were injected intra-articularly with anti-OSCAR antibodies, hOSCAR-Fc, or isotype-control hIgG1 (all 2 mg/kg, at which dose hOSCAR-Fc showed efficacy in the DMM model [14]) twice a week for 8 weeks starting 1 week after DMM surgery. Sham-operated mice also served as controls. The mice were euthanized 9 weeks after OA induction to model late-stage OA [32], and their knee joint tissues were subjected to histological and biochemical analyses. 

### 2.13. Histology, Immunohistochemistry, and TUNEL Fluorescence Microscopy of Joint Sections

At the end of the in vivo experiments, the knee joints of the mice were fixed in 10% formaldehyde at 4 °C for >48 h, decalcified in 0.5 M ethylenediaminetetraacetic acid in PBS (pH 7.4) for 14 days, and embedded in paraffin. The paraffin blocks were then cut into 5 μm-thick sections and stained with safranin-O. Articular cartilage destruction was scored with Osteoarthritis Research Society International (OARSI) grading (0–6), which is a standard OA-grading system [33]. Subchondral bone plate sclerosis and articular cartilage destruction were identified by safranin-O staining and measured using OsteoMeasureXP (OsteoMetrics, Inc., Atlanta, GA, USA), Adobe Photoshop (v19.1.3, San Jose, CA, USA), and an Olympus DP72 charge-coupled device camera (v2.1, Olympus Corporation, Tokyo, Japan). For immunohistochemistry, the knee-joint sections were incubated overnight at 4 °C with a primary antibody specific for OSCAR (Cat. # PA5-47171, Thermo Fisher Scientific, 1:100 dilution), OPG (Cat# sc8468, Santa Cruz Biotechnology, 1:200 dilution), or TRAIL (Cat# AF1121, R&D Systems, 1:200 dilution). Immunoactivity was detected by using a DAB peroxidase substrate detection kit (Vector Laboratories, Inc., Newark, CA, USA), and the nuclei were counterstained with hematoxylin. Articular chondrocyte apoptosis was determined by using a TUNEL assay kit (Millipore, Temecula, CA, USA). The specimens were visualized under a fluorescence microscope (v2.1, Olympus DP72 charge-coupled device camera, Olympus Corporation, Tokyo, Japan), and the apoptotic articular chondrocyte numbers in relation to the total cell numbers were determined. 

### 2.14. Quantitation and Statistical Analysis

All data were from at least four independent experiments. Groups were compared by two-way analysis of variance (ANOVA) followed by Sidak’s Multiple Comparisons test. All *p*-values are indicated in the figures. The error bars represent the standard error of the mean (S.E.M.). All graphs and statistical analyses were made by using GraphPad Prism (v8.1.2, GraphPad Software, Boston, MA, USA). 

## 3. Results

### 3.1. Generation of Anti-OSCAR Antibodies by Phage Display

Synthetic human single chain variable fragment (scFv) libraries [24,25] were panned in alternating rounds against human and mouse OSCAR extracellular domains that were fused to human IgG1 Fc (designated hOSCAR-Fc and mOSCAR-Fc). ELISA screening and sequencing of the third- and fourth-round output clones revealed seven unique scFv sequences that demonstrated species cross-reactivity. These clones were converted into hIgG1 by cloning the variable domains into pcIW3.3 vector (a pcDNA3.3-based mammalian expression vector that was constructed in-house by adding unique restriction sites, an artificial intron, a Kozak consensus sequence, a human serum albumin signal sequence, and the woodchuck hepatitis virus post-transcriptional response element). The ability of the converted IgGs to bind to both human and mouse OSCAR was then assessed by ELISA. Five of the seven clones retained their cross-species OSCAR-binding activity. However, while the IgGs of the remaining two clones, namely, D2 and D11, bound strongly to mOSCAR-Fc, they could no longer bind significantly to hOSCAR (Figure 1A). Nonetheless, they were included in subsequent in vitro functional assays with murine cells, with the assumption that their hOSCAR-binding activity could be rescued by further engineering.

### 3.2. In Vitro Functional Characterization of Anti-OSCAR Antibodies

Our previous study showed that chondrocytes express OSCAR when they are treated with collagen [14]. Moreover, such OSCAR expression induces the chondrocytes to express OA-associated ECM-degrading enzymes such as MMP3 and MMP13. This phenomenon was used to identify which of the seven anti-OSCAR antibody clones could inhibit the pro-catabolic activities of OSCAR. Thus, the murine ATDC5 cell line, which differentiates into chondrocytes in vitro [34,35], was cultured with each antibody and then treated with collagen. ATDC5 expression of *Oscar*, *Mmp3*, and *Mmp13* was then examined by quantitative RT-PCR. The expression of *Epas1* was also assessed: *Epas1* (endothelial PAS domain-containing protein 1) encodes hypoxia-inducible factor-1α, which is known to upregulate MMP expression in OA (Figure 1B). The positive control inhibitor used in this experiment was hOSCAR-Fc, which served as a decoy receptor. Compared to the isotype control antibody (hIgG1), hOSCAR-Fc and five of the seven anti-OSCAR antibodies (B4, B12, C11, D2, and D11) blocked the collagen-induced downstream signaling activity of OSCAR in ATDC5 cells. By contrast, the A1 and H10 antibodies enhanced OSCAR expression and activity and were consequently excluded from the study (Figure 1C). We also excluded the B12 and C11 antibodies because of their low expression. The D2 antibody was also excluded due to stability issues that caused antibody aggregation upon storage at 4 °C for several days. Consequently, only the B4 and D11 antibodies were selected for testing in further experiments.

To confirm that B4 and D11 can inhibit the OSCAR functions in chondrocytes, we exploited another recent finding. Specifically, when primary articular chondrocytes are cultured with the OSC peptide, which bears the minimal triple helix motif recognized by OSCAR [19], and then with IL-1β, they start expressing not only Oscar but also Tnfsf10. The latter gene encodes TRAIL, and its upregulation by OSC promotes chondrocyte apoptosis [14]. Thus, primary murine chondrocytes were cultured with OSC or the GPP10 control peptide and either B4, D11, hOSCAR-Fc, or hIgG, followed by IL-1β. Indeed, OSC + IL-1β increased OSCAR and TRAIL expression, and this was effectively blocked by B4, D11, and OSCAR-Fc (Appendix A).

OSCAR is best known for its ability to act as a co-stimulatory receptor of osteoclastogenesis [16,20]. Thus, to further confirm that B4 and D11 can inhibit OSCAR functions, we also tested them in the osteoclast differentiation assay. This assay involves co-culturing osteoclast progenitor cells with primary osteoblasts, which induces the progenitors to differentiate into mature osteoclasts [28]. The hOSCAR-Fc positive control, B4, and D11 all inhibited osteoclast differentiation in a dose-dependent manner (Appendix A). 

### 3.3. Affinity Maturation of Anti-OSCAR Antibodies

Given that D11 lost its ability to bind to hOSCAR when it was converted to IgG, we further optimized both the B4 and D11 antibodies by affinity maturation. The light chain shuffling approach was employed first. Thus, the lambda and kappa light chain variable domain (VL) repertoires of the OPAL-S scFv library [25] were amplified by PCR and then assembled with the heavy chain variable domain (VH) of B4 by overlap extension PCR, and the resulting PCR products were ligated to pComb3X phagemid vector to construct VL-shuffled scFv phagemid libraries that contained 3.3 × 10^7^ kappa light chain and 1.9×10^7^ lambda light chain transformants (Figure 2A). These libraries were then panned against OSCAR to yield B4 variants with improved affinity. However, the same approach failed to produce D11-derived clones with improved affinity. Therefore, we employed NNK-walking randomization of CDR-H3 for D11, in which two consecutive codons of the CDR-H3 sequence of D11 were randomized in an overlapping manner by using the NNK degenerate codon (N = any of the four deoxynucleotides; K = G or T) (Figure 2A) The resulting small libraries (each of which contained 400 unique amino acid sequences; 1.4 × 10^5^–1.8 × 10^6^ transformants) were panned individually against OSCAR.

Three rounds of panning against hOSCAR-Fc were performed with increasing stringency. Thus, 1 µg/mL hOSCAR-Fc and five washes with PBST were used for the first two rounds, and 0.2 µg/mL hOSCAR-Fc, eight normal washes, and two extended washes involving 30 min incubation were used for the third round. The output clones were then screened by ELISA, and the binders were sequenced. Three unique binder sequences from the light chain shuffling of B4 and six from the CDR-H3 randomization of D11 were identified and subjected to further characterization.

Two of the B4 variants had kappa light chains (B4K-C3 and B4K-C11), whereas the parental B4 clone and B4L-E2, the third B4 variant, had a lambda light chain. The CDR-L1s and CDR-L3s of B4K-C3 and B4K-C11 belong to the VK1 family, whereas the CDR-L2s were from the VK3 family. This reflects the fact that the light chain repertoire used in the chain shuffling was from a synthetic scFv library with a single scaffold and diversified CDRs [25] and implies that a single variable domain framework can accommodate CDRs of different germline family origins. The CDR-L1, L2, and L3 of the B4L-E2 variant were from the VL3, VL1, and VL3 families, respectively, compared to VL1, VL3, and VL1, respectively for the parental B4. The light chain CDRs of the B4 variants showed no significant similarity to the parental sequences, in contrast to the affinity-matured variants typically obtained by other approaches such as focused CDR randomization [36] or error-prone PCR [37], which tend to yield similar CDR sequences.

Of the six D11 variants obtained by CDR-H3 randomization, three were from NNK randomization of residues H97–98 (Kabat numbering), and one each was from the randomization of residues H98–99, H99–100, or H100–101. Randomization of residues H95 and H96 did not yield any OSCAR binders, which suggests that these residues are critical for the binding of D11 to OSCAR. Interestingly, the CDR-H3s of the variant clones obtained from the randomization of residues H98-99, H99-100, and H100-101 (D11-B9, D11-F8, and D11-C10, respectively) were shorter than the CDR-H3 of the parental clone by one or two amino acids (Table 2). This was presumably caused by oligonucleotide synthesis errors. The isolation of these rare deletion mutants suggests that varying CDR length, as well as CDR sequence, can promote in vitro antibody affinity maturation.

### 3.4. Affinities of the Optimized Antibodies for Human and Mouse OSCAR-Fc

B4 and D11 variant clones were expressed and purified from *E. coli* in scFv form and tested by serial dilution ELISA, and clones with markedly enhanced binding activity were reformatted to IgG. The affinities of B4, D11, and their variants in IgG format for human and mouse OSCAR-Fc were then assessed by serial dilution ELISA against hOSCAR-Fc and mOSCAR-Fc (Figure 2B and Table 3). As noted above (Figure 1A), D11 largely lost its ability to bind to human OSCAR after it was converted to IgG: Its EC_50_ for hOSCAR was >1000 nM, whereas its affinity for mOSCAR was much higher (151 nM). All B4 and D11 variants demonstrated higher affinity than their parents with EC_50_ values in the nanomolar range (0.93–36.8 nM). Notably, the mutations in the CDR-H3 of D11 restored its ability to bind to hOSCAR: The EC_50_ value of D11-B9 for hOSCAR was 36.8 nM (vs. >1000 nM for D11).

The binding kinetics of B4L-E2, B4K-C11, D11-B9, and their parent antibodies were analyzed by biolayer interferometry. The variants had sub-nanomolar *K*_D_ values for both mOSCAR and hOSCAR (0.088–0.41 nM), whereas the parents had nanomolar *K*_D_ values (2.87–5.61 nM) (Figure 2C and Table 4). Interestingly, despite the nanomolar *K*_D_ values for D11, the response of D11 IgG (in wavelength shift) for hOSCAR-Fc was much lower than for mOSCAR-Fc. This is consistent with the weaker binding of D11 to hOSCAR-Fc in the ELISA (Figure 1A and Figure 2B and Table 3). It is possible that the human antigen used in this study bore structural heterogeneity to which D11 was sensitive. However, the exact binding mechanism was not elucidated in this study.

### 3.5. In Vivo Efficacy of the Affinity-Matured Anti-OSCAR Antibodies

The DMM OA model mimics the OA that is commonly induced in humans by clinical meniscal injury [30,31]. It is a commonly used model of OA and thus was used to test the therapeutic efficacy of B4K-C11, B4L-E2, D11-B9, and their parent antibodies. The anti-OSCAR antibodies or the positive control hOSCAR-Fc were injected intra-articularly twice a week for 8 weeks, starting 1 week after the DMM surgery. After euthanizing the mice at 9 weeks, the joints were subjected to histological analysis. The parental B4 and D11 antibodies inhibited the cartilage degradation with comparable efficacy to hOSCAR-Fc, as indicated by the OARSI grade [33] (Appendix A). The B4K-C11, B4L-E2, and D11-B9 antibodies also significantly reduced the OARSI grades relative to the isotype control group (Figure 3A,C). All antibodies also markedly reduced other DMM-induced OA signs, namely, subchondral bone plate sclerosis and loss of hyaline cartilage (Figure 3B,C). Notably, D11 and its D11-B9 variant inhibited OA more effectively than B4 and its variants. Moreover, immunohistochemical analysis of the cartilage tissues confirmed that antibody-mediated inhibition of OSCAR downregulated the OA-associated TRAIL expression by articular chondrocytes while upregulating their OPG expression (Appendix A). Furthermore, TUNEL assays of the cartilage tissues showed that the anti-OSCAR antibodies significantly reduced chondrocyte death. In particular, D11 and D11-B9 completely suppressed OA chondrocyte apoptosis (Figure 4 and Appendix A). Collectively, these results show that the anti-OSCAR antibodies significantly inhibit OSCAR functions, particularly its ability to promote chondrocyte apoptosis and thereby induce OA.

## 4. Discussion

Our previous study showed that blocking OSCAR with intra-articular injections of a soluble protein containing the extracellular domains of OSCAR (hOSCAR-Fc) reduces cartilage destruction in OA [14]. However, hOSCAR-Fc exerts these effects by binding to collagen, which is ubiquitous and highly abundant in the body. Thus, the risk of undesirable side effects may be high. By contrast, monoclonal antibodies that bind with high specificity and affinity to OSCAR and thereby block its pro-chondrolytic signaling are likely to constitute a safer approach. This approach to drug development is also readily facilitated by well-established technologies for engineering, optimizing, and producing monoclonal antibodies, which have made monoclonal antibodies particularly suited to drug development. Thus, in the present study, we sought to develop OSCAR-neutralizing antibodies.

The extracellular Ig-like 1 and 2 domains of the human and mouse OSCAR proteins show moderate sequence identity (77.6%). Since species cross-reactivity is desirable for the preclinical and clinical development of antibody-based drugs, we panned scFv libraries against both human and mouse OSCAR in alternating rounds. Binding to mouse OSCAR is necessary for experiments, including cell-based assays, efficacy tests in animal models, and toxicological and pharmacokinetics studies, while affinity to human OSCAR is required for clinical applications. This led to seven scFv clones that recognized both proteins. Once they were converted to IgG, however, two of the clones (D2 and D11) largely lost their ability to bind to hOSCAR, although they could still strongly recognize mOSCAR. Nonetheless, we continued to investigate these antibodies because we assumed that the lost binding activity could be regained by affinity maturation. Indeed, when D11 was affinity-matured (D2 was later abandoned due to stability issues), its variants regained the ability to recognize hOSCAR.

After identifying the seven OSCAR-binding antibodies, we asked whether they could neutralize the cellular activities of OSCAR. Thus, we assessed their effect on the collagen-induced expression of OA-marker genes (*Mmp3*, *Mmp13*, and *Epas1*) by the chondrogenic ATDC5 cell line. This cell line is of murine origin, thus allowing D11 and D2 to be tested as well. Five of the seven antibodies inhibited OSCAR-mediated expression of the OA marker genes, which suggested they might also have in vivo efficacy against OA. While these five antibodies exhibited comparable in vitro efficacies, only B4 and D11 were selected for further development based on their higher expression levels and better stability. They also effectively inhibited two other known cellular OSCAR functions, namely, (i) OSC peptide + IL-1β-induced expression of the OSCAR and TRAIL genes in murine articular chondrocytes [14,19] and (ii) differentiation of murine osteoclast progenitors into mature osteoclasts [15,16]. 

B4 and D11 were subjected to affinity maturation in order to enhance their efficacies and also to rescue the binding activity of D11 to hOSCAR, which was lost after its conversion to IgG. Light chain shuffling was attempted first and proved successful with B4 but not with D11. Consequently, D11 was subjected to focused CDR-H3 randomization. The B4 parent antibody already had nanomolar dissociation constants for hOSCAR and mOSCAR, and affinity maturation improved this to sub-nanomolar levels. This was also observed with D11 for mOSCAR, and notably, the affinity maturation of D11 completely restored its response to hOSCAR-Fc: The EC50 improved from >1000 nM for D11 to 0.93 nM for the D11-B9 variant. Interestingly, the CDR-H3 of D11-B9 not only bore amino acid substitutions but was also one amino acid shorter than its parental sequence. Both of these changes probably significantly altered the CDR conformation [38], thereby markedly improving the affinity of the antibody. The fact that the light chain shuffling approach improved the affinity of B4 but not D11, coupled with the considerable difference in the CDR-H3 sequence between D11-B9 and D11, suggests that the target binding activity of D11 relies comparatively more on VL to compensate for the suboptimal CDR-H3 sequence.

In the in-vivo OA model, B4, D11, and their affinity-matured variants effectively blocked OA pathogenesis. Particularly, D11 and its variant D11-B9 performed better than B4 and its variants. Given the overall similar affinities of these antibodies, it is likely that this efficacy difference reflects differences between D11 and B4 in the epitopes they recognize or their mode of binding. Notably, however, the in vivo efficacies of the parent antibodies did not differ significantly from those of the affinity-matured clones despite the >10-fold lower *K*_D_ values of the latter. We speculate that the local concentration of intra-articularly injected antibody was high enough to saturate the OSCAR molecules on the chondrocytes, even if the antibody (e.g., B4 and D11) only had moderate affinity. Affinity maturation of D11 was still necessary to rescue the affinity toward human OSCAR (see above). Higher affinity antibodies might also prove more efficacious at a lower dose or with other routes of administration. 

It is noted that the hOSCAR-Fc decoy receptor, and not mOSCAR-Fc, was used as a positive control in the cell-based assays as well as the animal studies, even though a murine chondrocyte cell line (ATDC5) and a mouse disease model, respectively, were used in these experiments. This is because hOSCAR-Fc was already successfully evaluated in these settings [14] with potential clinical applications in mind, for which its human-derived sequence would be advantageous with lower immunogenicity.

While the antibody-targeting of OSCAR is a promising strategy for the treatment of OA, the approach is not without limitations. OA is a chronic, non-lethal disease, and an exceptional long-term safety profile is required for OA therapy. Because OSCAR is also expressed in osteoclasts, monocytes, and other immune and non-immune cell types [17,39], targeting and neutralizing OSCAR might result in unexpected and undesirable on-target, off-tissue effects. Also, as with most other OA therapies, anti-OSCAR therapy is unlikely to reverse the extensive cartilage damage of late-stage disease. Further studies are needed to better understand the advantages and limitations of OSCAR blockade and its mode of action and to develop anti-OSCAR agents as OA therapy. 

## 5. Conclusions

This study showed that OSCAR-neutralizing antibodies with human/mouse cross-species reactivity not only downregulated the in vitro expression of OSCAR, ECM-degrading enzymes, and TRAIL in chondrocytes they also reduced chondrocyte TRAIL expression and apoptosis, cartilage destruction, and other OA-related signs in an animal model of surgically induced OA. This is consistent with the ability of the hOSCAR-Fc decoy receptor to block OA [14]. Thus, these anti-OSCAR antibodies are promising and warrant further studies as candidate DMOADs.

## Figures and Tables

**Figure 1 biomedicines-11-02844-f001:**
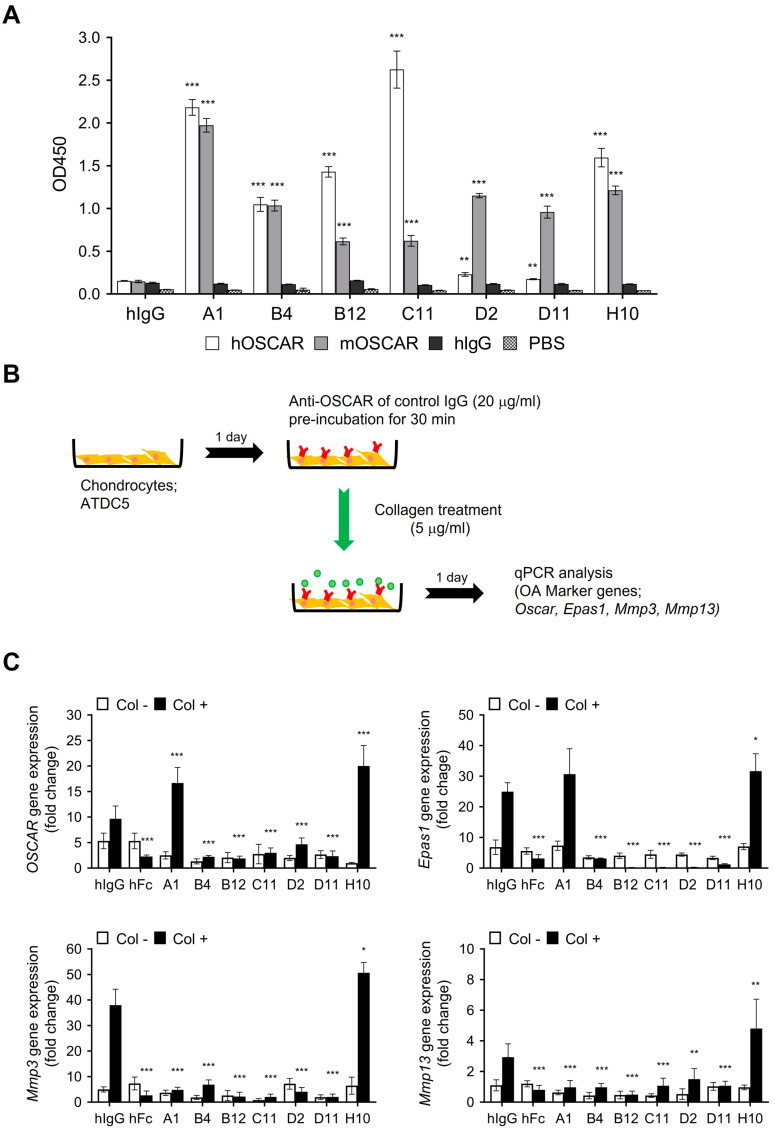
Ability of candidate anti-OSCAR Abs to regulate OSCAR expression and pro-catabolic downstream signaling activity in chondrocytes. (**A**) Ability of seven anti-OSCAR hIgGs to bind human and mouse OSCAR-Fc on ELISA. (**B**) Schematic depiction of the assay used to identify antibodies that could inhibit the collagen-I-induced pro-catabolic downstream signaling of OSCAR in chondrocytes. (**C**) The *Oscar*, *Epas1*, *Mmp3*, and *Mmp13* mRNA levels in the ATDC5 cell line were determined by using quantitative RT-PCR. hFc: the human OSCAR-Fc fusion protein, which served as the positive control inhibitor. hIgG: the isotype control, which served as the negative control. Error bars represent mean ± S.E.M. Two-way ANOVA followed by Sidak’s Multiple Comparison test was performed. * *p* < 0.05, ** *p* < 0.01, *** *p* < 0.001 when the anti-OSCAR antibody- or hOSCAR-Fc-treated group was compared to the hIgG negative control group.

**Figure 2 biomedicines-11-02844-f002:**
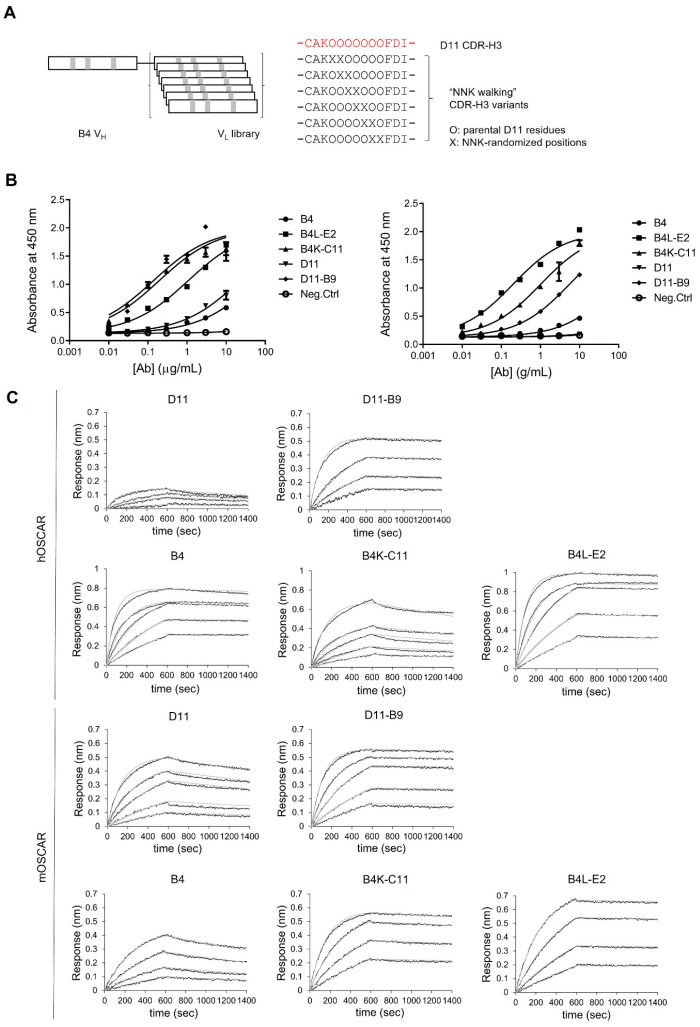
Affinity maturation of the B4 and D11 anti-OSCAR antibodies. (**A**) Schematic depiction of the methods used to diversify the B4 and D11 parental antibodies. B4 was diversified by light chain shuffling, while D11 was diversified by two-codon NNK walking of the CDR-H3 (parental D11 CDR-H3 sequence is shown in red). (**B**) Serial-dilution ELISA to determine the affinity of the B4 and D11 parent clones and their affinity-matured variants for mouse (left) and human (right) OSCAR-Fc. (**C**) Biolayer interferometry analysis of the binding kinetics of the anti-OSCAR antibodies.

**Figure 3 biomedicines-11-02844-f003:**
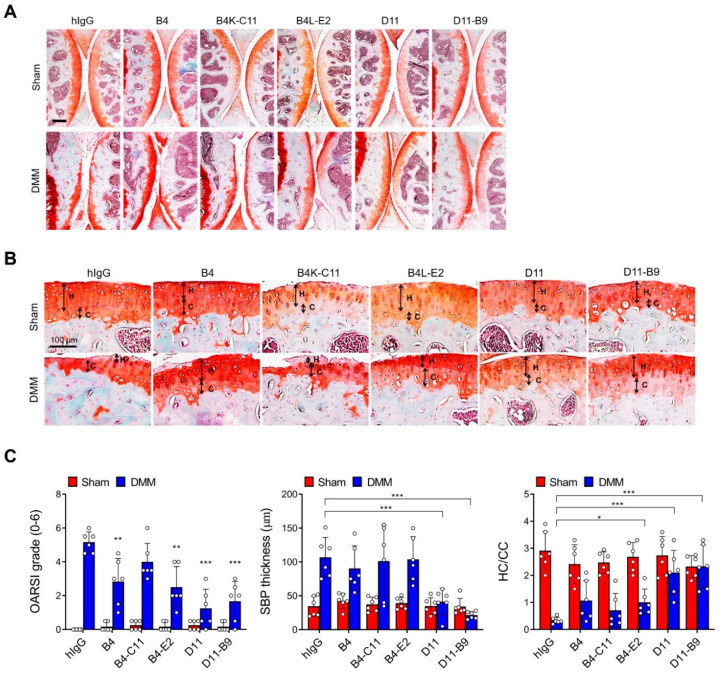
The anti-OSCAR B4 and D11 antibodies and their affinity-matured variants block OA in a murine model. Wild-type mice were subjected to DMM or sham surgery and then injected intra-articularly twice a week for 8 weeks with each anti-OSCAR antibody (*n* = 6 per group). The mice were sacrificed 9 weeks after surgery, and the articular cartilage in their joints was sectioned and stained with safranin-O to determine the OARSI grade, subchondral bone plate (SBP) thickness, and the ratio of hyaline cartilage (HC) to calcified cartilage (CC). (**A**) Representative images of the safranin-O-stained sections. (**B**) Representative magnified images showing the HC and CC. (**C**) Quantitative analysis of OARSI grade, subchondral bone plate thickness, and HC:CC ratio. Scale bars: 100 μm. Error bars represent mean ± S.E.M. Two-way ANOVA followed by Sidak’s Multiple Comparison test was conducted. * *p* < 0.05, ** *p* < 0.01, *** *p* < 0.001 when comparing the anti-OSCAR antibody-treated groups to the hIgG control group.

**Figure 4 biomedicines-11-02844-f004:**
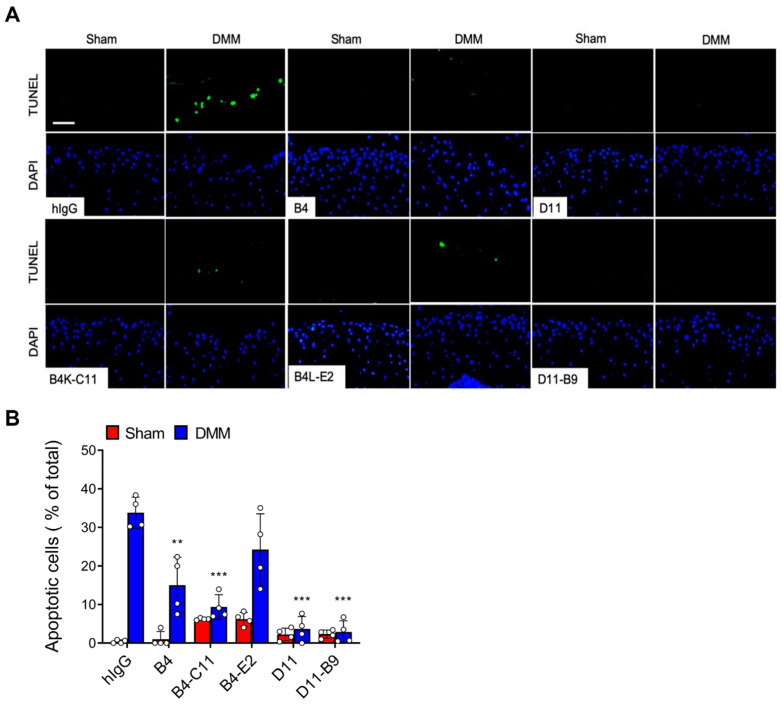
The anti-OSCAR antibodies ameliorate OA by blocking articular chondrocyte apoptosis. Wild-type mice were subjected to DMM or sham surgery and then injected intra-articularly twice a week for 8 weeks with each anti-OSCAR antibody (*n* = 6 per group). The mice were sacrificed 9 weeks after surgery, and the articular cartilage in their joints was sectioned. The apoptotic articular chondrocytes were detected and quantified by TUNEL assay. (**A**) Representative images. Scale bar: 50 μm. (**B**) Quantitative analysis of the apoptotic chondrocyte numbers. Error bars represent mean ± S.E.M. Two-way ANOVA followed by Sidak’s Multiple Comparison test was conducted. ** *p* < 0.01, *** *p* < 0.001 when anti-OSCAR antibody-treated groups were compared to the hIgG control group.

**Table 1 biomedicines-11-02844-t001:** Oligonucleotides used for real-time PCR analysis.

Gene	Strand	Primer Sequence	Origin
*Oscar*	Sense	5′-AGGGAAACCTCATCCGTTT-3′	Mouse
	Antisense	5′-TGCTGTGCCAATCACAAGTA-3′	
*Mmp3*	Sense	5′-TCCTGATGTTGGTGGCTTCAG-3′	Mouse
	Antisense	5′-TGTTTATTGTTGCTGCCCAT-3′	
*Mmp13*	Sense	5′-CCTTGAACGTCATCATCAGG-3′	Mouse
	Antisense	5′-TGTTTATTGTTGCTGCCCAT-3′	
*Epas1*	Sense	5′-CGAGAAGAACGACGTGGTGTTC-3′	Mouse
	Antisense	5′-GTGAAGGCGGGCAGGCTCC-3′	
*Tnfsf10*	Sense	5′-CCTCTCGGAAAGGGCATTC-3′	Mouse
	Antisense	5′-TCCTGCTCGATGACCAGCT-3′	
*Tnfrsf11b*	Sense	5′-CAGAGCGAAACACAGTTTG-3′	Mouse
	Antisense	5′-CACACAGGGTGACATCTATTC-3′	
*β-Actin*	Sense	5′-TGGAATCCTGTGGCATCCATGAAAC-3′	Mouse
	Antisense	5′-TAAAACGCAGCTCAGTAACAGTCCG-3′	

**Table 2 biomedicines-11-02844-t002:** CDR-H3 sequences of affinity-matured D11-variant clones.

Clone	CDR-H3 Sequence
D11 (parent)	AKAGFTGGHFDI
D11-3-B2	AKAGNWGGHFDI
D11-3-F2	AKAGNFGGHFDI
D11-3-F9	AKAGLFGGHFDI
D11-4-B9	AKAGIPG-HFDI
D11-5-F8	AKAG—FTHFDI
D11-6-C10	AKAG—FSMFDI

**Table 3 biomedicines-11-02844-t003:** EC50 values of the anti-OSCAR antibodies for binding to human and mouse OSCAR.

EC_50_ (nM)	D11	D11-B9	B4	B4K-C11	B4L-E2
hOSCAR	>1000	36.8	590	6.74	1.33
mOSCAR	151	0.93	414	1.30	7.20

**Table 4 biomedicines-11-02844-t004:** Binding kinetics of the anti-OSCAR antibodies, as determined by biolayer interferometry.

Ligand	Parameters	D11	D11-B9	B4	B4K-C11	B4L-E2
hOSCAR	*k*_on_ (M^−1^s^−1^)	9.13 × 10^4^	1.01 × 10^5^	8.99 × 10^4^	1.54 × 10^5^	1.38 × 10^5^
	*k*_off_ (s^−1^)	5.12 × 10^−4^	4.15 × 10^−5^	2.92 × 10^−4^	3.37 × 10^−5^	1.21 × 10^−5^
	*K*_D_ (nM)	5.61	0.41	3.25	0.22	0.088
mOSCAR	*k*_on_ (M^−1^s^−1^)	9.13 × 10^4^	9.13 × 10^5^	9.13 × 10^4^	9.13 × 10^5^	9.13 × 10^5^
	*k*_off_ (s^−1^)	2.54 × 10^−4^	2.59 × 10^−5^	3.89 × 10^−4^	6.79 × 10^−5^	2.47 × 10^−5^
	*K*_D_ (nM)	2.87	0.22	4.61	0.40	0.23

## Data Availability

The data presented in this study are available on request from the corresponding author. The data are not publicly available due to pending patents.

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
