# Peer review of "Development of Anti-OSCAR Antibodies for the Treatment of Osteoarthritis"

_biomedicines, 2023, doi:10.3390/biomedicines11102844_

Round 1

Reviewer 1 Report

Dear Authors and Reviewers!

Thank you so much for the opportunity to review the manuscript.

Osteoarthritis is a very important part of rheumatology, internal medicine and heriatrics. Millions people suffer from osteoarthritis, it the most frequent disease with joint involvement and walking problem. There are no specific treatment for this disease. The idea to use monoclonal antibodies in the treatment is very promising. The actuality is clear, the goals of the study corresponds with title, methods and results. The results are clear and are the basis for further studies and the following randomized trials. The discussion contain all modern literature.

Well-written and well-organized manuscript.

I only ask the Authors to add the Limitation subsection in the end of the Discussion.

Thank you so much!!!

Reviewer 2 Report

Development of anti-OSCAR antibodies for the treatment of osteoarthritis

The study is interesting. However, I have several comments for the authors.

Specific comments:

·         Introduction on OA should be improved. As the authors reported, OA is a whole joint disease involving all joint tissue including meniscus and infrapatellar fat pad other than cartilage, synovial membrane and subchondral bone. All the tissues should be mentioned.

·         The authors described that are chondrocytes are exposed continuously to external mechanical forces and play key roles in cartilage homeostasis because they produce both the cartilage ECM molecules and the metalloproteinases (MMPs) … The authors should also explain that chondrocytes undergo to biomechanical changes in OA (DOI: 10.3390/biomedicines11071942) and this is also true for cartilage and should be mentioned (DOI:10.3390/pr11041014 etc).

·         Lines 57-58: “Although articular chondrocytes usually do not express OSCAR, or do so only at low levels, its expression is strongly upregulated in OA.” References should be provided.

·         Lines 63-65: why did the authors inject soluble human OSCAR-Fc fusion protein and not a mouse OSCAR-Fc fusion?

·         At the end of the introduction, the aim/s of the study should be added rather than the results/conclusions.

·         Lines 76-77: the authors reported that primary articular chondrocytes were obtained from Institute of Cancer Research (ICR) mice. This part should be better explained. Did the authors buy mice chondrocytes? How were chondrocytes isolated? Passage?

·         Did the authors follow the ARRIVE guidelines for mice experiments?

·         Section 2.6: how did the authors analyze the data? Did the authors use the 2^ddct method?

·         Lines 130-131: Articular chondrocytes were isolated from the femoral condyles and tibial plateaus of 4–5-day-old ICR mice by digestion with 0.2% collagenase type II. Are these different chondrocytes from those described at lines 76-77?

·         Line 140: supplier of IL-1 beta should be added.

·         Section 2.8: cell media should be added. Did the authors characterize isolated BMMs and osteoblasts?

·         Tables and figures should be placed where they are cited for the first time (following the guidelines of the journal).

·         Lines 153-154: how many cells in total were counted?

·         Line 163: did the authors use a kit to isolate the PCR product from agarose gel?

·         Lines 192-193: How did the authors select the dose?

·         Line 194: why did the authors select to euthanize the mice 9 weeks after surgery?

·         Line 213: microscope used should be specified.

·         Lines 218: “The sample size for each experiment was not predetermined.” I suggest to delete this sentence.

·         Section 3.1:  considering that the authors performed all experiments on mice, it is unclear to me why the authors generated anti-OSCAR antibodies using also hOSCAR-Fc.

·         Gene names should be written in italics.

·         Lines 268-269: could the authors better explain the stability issue?

·         Figure 1a and 1c: statistical analysis is missing.

·         Supplementary figure 1 a: Why did the authors incubate chondrocytes with IL 1beta 5 ng/ml for 48h? Normally chondrocytes are incubated with Il1beta 10 ng/ml for 24h.

·         Supplementary figure 1 a: it is unclear if IL-1beta was added in all conditions except “none”.

·         Lines 288-289: this part should be better explained.

·         Figure 2: Bio-layer interferometry was not reported in the methods.

·         Line 340: “E.coli” should be written in italics.

·         Section 3.4: again, it is not clear to me why authors focused on both human and mouse OSCAR, considering that the paper is focused on mice and not humans. I noticed that this point was partly explained at the beginning of the discussion.

·         Could the authors explain why they inject hOSCAR-Fc and not mOSCAR-Fc as control?

·         The discussion seems a summary of the results. It should be improved.

·         Limitations of the study should be added.

Round 2

Reviewer 2 Report

No additional comments.